# Recognition of children's emotional facial expressions among mothers reporting a history of childhood maltreatment

Jessica Turgeon[1]*, Annie Bérubé[2], Caroline Blais[2], Annie Lemieux[2], Amélie Fournier[1]

1 Department of Psychology, University of Québec at Trois-Rivières, Trois-Rivières, Québec, Canada,
2 Department of Psychoeducation and Psychology, University of Québec in Outaouais, Gatineau, Québec, Canada

* jessica.turgeon@uqo.ca

**Data Availability Statement:** All relevant data are within the paper and its Supporting Information files.

## Abstract

Several studies have shown that child maltreatment is associated with both positive and negative effects on the recognition of facial emotions. Research has provided little evidence of a relation between maltreatment during childhood and young adults' ability to recognize facial displays of emotion in children, an essential skill for a sensitive parental response. In this study, we examined the consequences of different forms of maltreatment experienced in childhood on emotion recognition during parenthood. Participants included sixty-three mothers of children aged 2 to 5 years. Retrospective self-reports of childhood maltreatment were assessed using the short form of the Childhood Trauma Questionnaire (CTQ). Emotion recognition was measured using a morphed facial emotion identification task of all six basic emotions (anger, disgust, fear, happiness, sadness, and surprise). A Path Analysis via Structural Equation Model revealed that a history of physical abuse is related to a decreased ability to recognize both fear and sadness in children, whereas emotional abuse and sexual abuse are related to a decreased ability to recognize anger in children. In addition, emotional neglect is associated with an increased ability to recognize anger, whereas physical neglect is associated with less accuracy in recognizing happiness in children's facial emotional expressions. These findings have important clinical implications and expand current understanding of the consequences of childhood maltreatment on parents' ability to detect children's needs.

## Introduction

Parental sensitivity refers to a parent's ability to interpret child signals correctly and to offer an appropriate response [1]. This allows the establishment of a secure attachment relationship [1, 2] and promotes the healthy development of young children [3, 4]. Among other things, parental sensitivity is associated with positive socioemotional functioning and academic outcomes [5, 6]. Conversely, a misinterpretation of the child's needs or avoidance of caregiving responsibilities leads to important developmental consequences, as the child is unable to use

**Funding:** The study was funded by the Centre de recherche universitaire sur les jeunes et les familles (CRUJeF) https://www.crujef.ca/, (JT) and the Social Sciences and Humanities Research Council (SSHRC) https://www.sshrc-crsh.gc.ca/, 430-2016-00521, (AB). The funders had no role in study design, data collection and analysis, decision to publish, or preparation of the manuscript.

**Competing interests:** The authors have declared that no competing interests exist.

his caregiver as a secure base to explore his environment and seek contact in case of distress [1, 7, 8].

The inability to meet children's primary needs, causing either potential or actual harm, is referred to as child maltreatment [9]. A recent meta-analysis of the global prevalence of child maltreatment suggests prevalence rates between 12% and 36% for abuse (sexual, physical, emotional) and neglect (physical, emotional) [10]. In the United States, researchers have found that 37.4% of children will be reported to authorities for suspected abuse or neglect before the age of 18 [11]. Billions of dollars are invested annually in response to this social phenomenon [12]. From a developmental perspective, damaging effects in the cognitive, social, verbal, physical, and emotional spheres of the child have been reported in several studies. Among other things, child maltreatment is associated with insecure and disorganized attachment relationships [13], motor delays [14], learning difficulties [15, 16], internalized behavior problems [17], and externalized behavior problems such as aggressivity and opposition [18]. Long-term consequences include an increased risk of delinquency [19], substance use disorder [20, 21], and parenting difficulties [22].

One of the most documented risk factors for child maltreatment is the caregiver's own history of maltreatment during childhood [23, 24]. Parents who experienced abuse or neglect during their childhood are more likely to perpetrate maltreatment themselves [25–27]. Researchers have suggested that facial emotion recognition may have a key role in the intergenerational continuity of maltreatment [28].

## Emotions

In general, individuals are largely skilled at recognizing others' emotional expressions [29]. Studies have found that the ability to discriminate emotional facial expressions increases throughout development [30]. Richoz and colleagues [31] determined that the overall recognition ability of the six basic emotions (i.e. anger, disgust, fear, happiness, surprise, and sadness) peaks during adulthood, except happiness that is mastered at a younger age. Studies have shown that a child's living environment can influence the development of perceptual abilities. Children who have secure attachment relationships and more discussions about emotions are better at recognizing emotions [32]. Conversely, children who experience abuse or neglect differ from non-maltreated children in their ability to understand and distinguish emotional facial expressions [33, 34].

## Emotion recognition in maltreated children

Most studies examining the relationship between maltreatment and emotion recognition have been conducted by Dr. Seth Pollak. In 2000, Pollak and colleagues [34] found that neglected children, compared to physically abused and non-maltreated children, had more difficulty distinguishing emotional expressions as well as a tendency to generalize sadness to other negative emotions. Shipman and colleagues [35] also revealed that, compared to non-maltreated peers, neglected children had lower abilities in both emotion regulation and emotion understanding of negative emotions, such as anger and sadness. Other studies have found that children who are physically abused have an increased sensitivity to facial expressions of anger in comparison to other basic emotions [36, 37]. Several studies have found that maltreated children are better to identify expressions of anger [38–41] and fear [42] compared to controls. Conversely, studies have found that maltreated children require more sensory information to recognize facial expressions of sadness and are less accurate at processing happy faces compared to non-maltreated peers [37, 43]. Sexual abuse, on the other hand, is associated with lower understanding of emotions as well as deficits in emotion regulation [44]. Together, these findings suggest that maltreatment influences children's recognition and processing of emotions.

### Emotion recognition in adults reporting childhood maltreatment

Few studies have examined the effect of childhood maltreatment on the recognition of facial displays of emotions in adults, especially in parents. Gibb and colleagues [45] assessed undergraduate students' interpretation of facial emotions by exposing them to morphed facial stimuli at different levels of intensity. Their study revealed that participants reporting a history of childhood maltreatment had an increased sensitivity to anger. They perceived this expression when the emotion was presented at lower intensity. Similarly, English, and colleagues [46] reported hypervigilance to fear in women who experienced emotional maltreatment as children when completing a facial emotion recognition task requiring a high cognitive load. Another study conducted by Germine and colleagues [47] found no significant association between adults' emotion discrimination abilities and childhood experiences of maltreatment. However, rather than assessing abuse and neglect alone, the authors included many other adversities such as exposure to parental substance abuse, mental illness or criminal behavior.

The relationship between childhood maltreatment and facial emotion discrimination in adults is thus an emerging topic. Moreover, very few studies have examined this relationship using child facial expressions. Arteche and colleagues [48] investigated how postnatal depression and anxiety affects the way mothers process infant emotions and found that mothers were better at recognizing happy faces compared to sad faces when the expressions were at lower levels of intensity. Dayton and colleagues [49] documented perceptions of facial emotions among a sample of pregnant mothers, some of whom experienced childhood maltreatment and/or domestic violence. They found that mothers who experienced both forms of maltreatment perceived ambiguous infant facial expressions more negatively than other mothers.

Altogether, these studies have attempted to investigate the link between childhood maltreatment and emotion recognition in adulthood. However, very few have looked at parents' recognition of child facial expressions. Most protocols have used infant pictures, although children won't express the full range of the six basic emotions before the age of two [50]. In a previous study, we found that mothers who had higher scores on the Childhood Trauma Questionnaire (CTQ) scales had lower performance scores on the emotion recognition task [51]. Overall scores were created and used in analyses, for both childhood maltreatment experiences and emotion recognition performances. The current study seeks to further explore these results, by drawing conclusions about the differential effects of five maltreatment subtypes on mothers' ability to recognize six emotions expressed on child faces. To our knowledge, no studies have examined associations between the processing of the six basic facial emotional expressions and the five most documented forms of maltreatment (physical, emotional and sexual abuse, physical and emotional neglect). Thus, the main objective of this study is to examine facial expression recognition ability using child facial expressions of emotions among mothers reporting a history of child maltreatment.

## Materials and methods

The Ethical Committee of the University of Québec in Outaouais approved the present study (UQO CER #2518-B). The preliminary analyses for this work were the subject of a master thesis by the first author, which has been accepted and archived (Turgeon, 2019; http://di.uqo.ca/id/eprint/1125). All tasks were performed according to the university's guidelines and regulations. Written informed consent was obtained from all subjects prior to their participation.

### Participants

Participation in the study was voluntary. Participants were recruited from two community organizations in the Outaouais region, as well as from advertising on Facebook and

recruitment posters displayed on university walls. Different recruitment environments were selected to ensure considerable variability in individual exposure to maltreatment. Sixty-three mothers of a biological 2 to 5-year-old child participated in the study. Mothers' mean age was 32 years (S.D. = 5.29, range = 22 to 45). More than half of subjects (53%) reported a family income of less than 24, 000$ a year. Most mothers had completed post-secondary education (53%). However, 42% reported being either unemployed or a stay-at-home parent. Finally, most mothers were of White-European descent (80%) and had either two (44%) or three children (21%) in their care.

Although path analysis is a relatively simple model, we will respect Rex Kline's statistical power criteria [52]. Kline suggests that there are ideally 15 subjects per estimator, 10 is common practice, and at least 5 subjects is required for a simple model. With a sample size of sixty-three, a maximum of 12 estimators will be tested in the final model.

## Procedure

Subjects were met by a research assistant in the community organization they attended or at the University for families who were recruited through Facebook and advertising posters. Prior to performing tasks, all subjects completed an initial assessment of socio-demographic information. Mothers then performed a computerized task to assess their ability to categorize facial expressions of basic emotions. Subsequently, the research assistant administered a questionnaire to mothers to assess adverse childhood experiences.

## Measures

**Adverse childhood experiences.** The French brief screening version of the CTQ developed by Paquette and colleagues [53] was used to assess experiences of childhood abuse or neglect. It is a self-administered and self-reported questionnaire and consists of 28 questions on a 5-point Likert-type scale. For each item, mothers must indicate if the affirmation is 1 (never true), 2 (rarely true), 3 (sometimes true), 4 (often true) or 5 (very often true) [54]. The items are categorized into five scales: emotional abuse, physical abuse, sexual abuse, emotional neglect, and physical neglect. Scores on each scale can range from 5 to 25, the latter being the most severe. The brief screening version of the CTQ has demonstrated excellent internal consistency (Cronbach's alpha varying between 0.79 and 0.94), as well as excellent temporal stability (varying between 0.76 and 0.96) [52].

**Recognition of children's facial expressions of emotion.** To assess mothers' ability to recognize facial expressions of emotions, we used a computerized task resembling the Facial Expression Megamix Task, which is used to reveal deficits in emotion recognition [55]. Faces from two children of White-European descent were selected in the Child Affective Facial Expression set (CAFE) database [56]. Mothers were exposed to different combinations of emotional expressions, morphed together using *FantaMorph* software. A total of 15 combinations of expressions was created, each including 5 levels of intensity (20%, 35%, 50%, 65%, 80). The entire task consisted of 450 trials, divided into three blocks, and presented in random order. Mothers were instructed to identify which of the six basic emotions (anger, disgust, fear, happiness, sadness, surprise) was dominant in each trial. A research assistant entered the mother's response by pressing the corresponding key on the computer's keyboard. To reduce any possible bias, there was no communication between the research assistant and the parent during the task, and photographs were presented in grayscale only. To calculate mothers' performance, arcsine-transformed unbiased hit rates for each emotion were computed [57]. This measure was designed to account for potential response biases in emotion recognition tasks, such as if the participant uses one specific emotion category more frequently than the others or chooses

the correct category by chance. This task is identical to the one used in a previously published article [51].

## Statistical analyses

First, pairwise comparisons were computed between each pair of emotions to compare the level of ability across the sixty-three mothers. Bonferroni correction for multiple tests was applied and a *p*-value = .003 was used to assess significant results between the comparisons. Second, a confusion matrix was constructed to measure the percentage of time an emotion presented at 50% or more in an image was confused with another emotion. This matrix gives information on participants' interpretive biases. For example, a score of .3 on the intersection between anger (horizontally) and disgust (vertically) would mean that participants confuse anger with disgust 30% of the time when anger is presented at 50% or more in the picture.

Third, descriptive and univariate correlational analyses were performed for all variables involved in the analysis. Finally, a path model via Structural Equation Modeling (SEM) was performed using MPlus 7.4 to assess the link between maltreatment and emotion recognition, while controlling for ethnicity, mother's education, and the number of children in the family. The full model with all estimate paths was not tested because of the relatively small sample size in the current study and the risk for multicollinearity. Only univariate correlations with *p*-values smaller than .10 were integrated into the first tested model (control variables and types of maltreatment with emotion recognition). This criterion is based on the work of Bendel & Afifi [58] and Mickey & Greenland [59] who show that a *p*-value cut-off of .05 often fails to identify important variables. We used a backward elimination strategy to reach a parsimonious model. Each non-significant (*p*-value >.10) relationship was eliminated from the model in a decreasing order based on the *p*-value. At each elimination step, the modification indices were verified to ensure that a previously removed link did not become significant (*p*-value < .10). Hu and Bentler's [60] statistical fits recommendations were used to validate the final model (non-significant Chi-Square, root-mean-square error of approximation (RMSEA) < .05; comparative fit index (CFI) >.95). Due to the small sample size, an examination of the residual values was made to ensure good validation of the model. Small residual values (close to zero) represent a good adjustment of the model.

## Results

### Paired *t*-tests

Controlling for multiple tests (Bonferroni corrected critical *p*-value = .003), mean accuracy was significantly lower for disgust and fear compared with all other emotions (*p* < .001), apart from the non-significant relationship between disgust and fear (*p* = .028). Contrarily, accuracy was significantly higher for happiness compared with all other emotions. Anger was the second emotion with the highest accuracy scores, followed by sadness and surprise, respectively. A full list of pair-tested comparisons is presented in Table 1. Mean accuracy and standard deviation for each emotion are presented in Fig 1.

### Confusion matrix

Table 2 shows patterns in the confusions that participants made during the emotion recognition task. Our data revealed that the largest misinterpretation in recognition made by participants was fear being mistaken for surprise (50%). The inverse error (surprise being interpreted as fear) also occurred at 24%. Moreover, mothers most often confuse disgust for anger (32%)

**Table 1. Results of paired-samples *t*-tests for each combination of emotions (n = 63).**

| | t(62) | Cohen d | 95% CI (low; high) | p |
|---|---|---|---|---|
| Anger-Disgust | -25.44 | -3.20 | -(0.47; -0.55) | < .001 |
| Anger-Fear | -21.68 | -2.73 | -(0.40; -0.48) | < .001 |
| Anger-Happiness | -17.18 | -2.16 | (-0.30; -0.24) | < .001 |
| Anger-Sadness | -07.61 | -0.96 | -(0.09; -0.15) | < .001 |
| Anger-Surprise | -14.79 | -1.86 | -(0.21; -0.27) | < .001 |
| Disgust-Fear | 0–2.25 | -0.28 | (-0.12; -0.01) | .028 |
| Disgust-Happiness | -32.52 | -4.10 | (-0.83; -0.73) | < .001 |
| Disgust-Sadness | -16.75 | -2.11 | (-0.43; -0.34) | < .001 |
| Disgust-Surprise | -10.31 | -1.30 | (-0.32; -0.21) | < .001 |
| Fear-Happiness | -30.33 | -3.82 | (-0.76; -0.67) | < .001 |
| Fear-Sadness | -13.49 | -1.70 | (-0.37; -0.28) | < .001 |
| Fear-Surprise | 0–8.81 | -1.11 | (-0.25; -0.16) | < .001 |
| Happiness-Sadness | -22.36 | -2.82 | -(0.36; -0.43) | < .001 |
| Happiness-Surprise | -27.77 | -3.50 | -(0.48; -0.55) | < .001 |
| Sadness-Surprise | -05.54 | -0.70 | -(0.08; -0.17) | < .001 |

*Note*. Bonferroni corrected critical *p*-value = 0.003.

CI, confidence interval (lower and upper bounds of the mean difference).

and anger for disgust (14%). A tendency to confuse disgust and sadness (26%) was also detected, as well as a confusion between sadness and fear (14%).

## Descriptive and correlational analyses

Table 3 depicts the mean, standard deviation, minimum, maximum, and distribution (skewness and kurtosis) for each variable involved in the analysis, as well as the bivariate correlations across all variables. Bivariate correlations revealed a negative significant association between the recognition of anger and the three types of abuse (physical, emotional, and sexual). The more severe the abuse, the less accurate mothers were at correctly recognizing the emotion

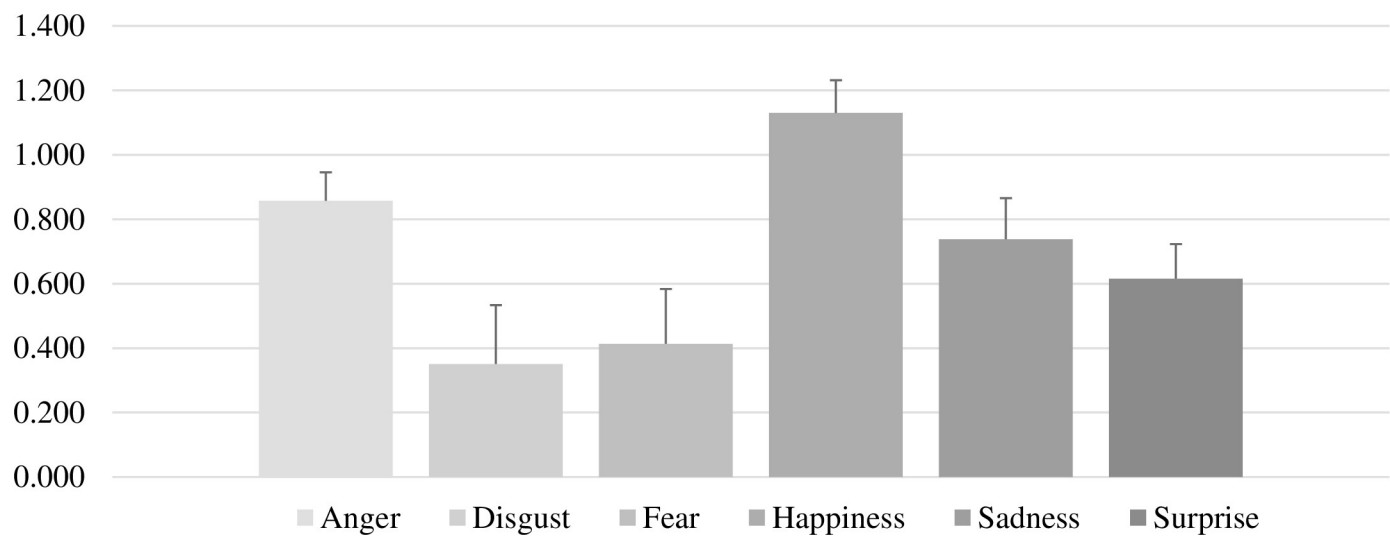

**Fig 1. Mean accuracy and standard deviation for each emotion.**

**Table 2. Confusion matrix using total performances (hit rates) on the emotion task (n = 63).**

|  | Anger | Disgust | Fear | Happiness | Sadness | Surprise |
|---|---|---|---|---|---|---|
| Anger | **0.82** | 0.14 | 0.01 | 0.01 | 0.01 | 0.01 |
| Disgust | 0.32 | **0.29** | 0.05 | 0.05 | 0.26 | 0.03 |
| Fear | 0.02 | 0.07 | **0.37** | 0.01 | 0.03 | 0.50 |
| Happiness | 0.01 | 0.02 | 0.00 | **0.93** | 0.00 | 0.03 |
| Sadness | 0.01 | 0.11 | 0.14 | 0.03 | **0.68** | 0.03 |
| Surprise | 0.00 | 0.02 | 0.24 | 0.05 | 0.04 | **0.64** |

when it was presented at 50% or more in the stimulus. The same relationship was found for fear and all types of abuse, as well as emotional neglect. Furthermore, physical neglect was negatively correlated with the recognition of happiness. The more mothers reported having a severe history of physical neglect, the less they were accurate at recognizing children's facial expressions of happiness (see Table 3). Finally, ethnicity was negatively correlated with the recognition of sadness, mother's education was positively correlated with anger, disgust, and fear, and the number of children in the family was significantly associated with less ability in fear recognition. These significant associations were included in the path model.

**Table 3. Bivariate correlations, means, and standard deviations among variables in the path model (n = 63).**

| Variable | 1. | 2. | 3. | 4. | 5. | 6. | 7. | 8. | 9. | 10. | 11. | 12. | 13. | 14. |
|---|---|---|---|---|---|---|---|---|---|---|---|---|---|---|
| 1. Ethnicity (0 = White-European descent, 1 = other) | - | -.26*** | -.15*** | -.09* | -.14 | .06 | -.03 | -.26* | -.03 | -.12 | -.09 | -.02 | -.10 | -.00 |
| 2. Mother's education |  | - | -.26*** | -.29** | .34** | .36** | -.02 | -.13 | -.01 | -.39** | -.36** | -.28* | -.41** | -.40** |
| 3. Number of children |  |  | - | -.15* | -.01 | -.35** | -.09 | -.05 | -.08 | -.33** | -.26* | -.28* | .38** | .37** |
| 4. Anger (UAT) |  |  |  | - | .50*** | .34** | -.12 | -.38** | -.13 | -.32** | -.34** | -.29* | -.17 | -.18 |
| 5. Disgust (UAT) |  |  |  |  | - | .21† | -.20 | -.34** | -.09 | -.16 | -.05 | -.17 | -.11 | -.10 |
| 6. Fear (UAT) |  |  |  |  |  | - | -.12 | -.20 | -.20 | -.36** | -.28* | -.21† | -.34** | -.20 |
| 7. Happiness (UAT) |  |  |  |  |  |  | - | -.27* | -.01 | -.05 | -.20 | -.17 | -.20 | -.25* |
| 8. Sadness (UAT) |  |  |  |  |  |  |  | - | -.12 | -.17 | -.12 | -.14 | -.07 | -.05 |
| 9. Surprise (UAT) |  |  |  |  |  |  |  |  | - | -.06 | -.02 | -.04 | -.07 | -.07 |
| 10. Physical abuse |  |  |  |  |  |  |  |  |  | - | -.72*** | -.72*** | -.69*** | -.65*** |
| 11. Emotional abuse |  |  |  |  |  |  |  |  |  |  | - | -.52*** | -.78*** | -.65*** |
| 12. Sexual abuse |  |  |  |  |  |  |  |  |  |  |  | - | -.54*** | -.62*** |
| 13. Emotional neglect |  |  |  |  |  |  |  |  |  |  |  |  | - | -.72*** |
| 14. Physical neglect |  |  |  |  |  |  |  |  |  |  |  |  |  | - |
| Mean | 0.16 | 3.21 | 2.48 | 0.86 | 0.35 | 0.41 | 1.13 | 0.74 | 0.62 | 7.19 | 9.44 | 6.98 | 9.83 | 7.67 |
| SD | 0.37 | 1.69 | 1.09 | 0.09 | 0.18 | 0.17 | 0.10 | 0.13 | 0.11 | 4.71 | 5.84 | 4.40 | 4.86 | 3.86 |
| Min | 0.00 | 1.00 | 1.00 | 0.65 | 0.03 | 0.04 | 0.92 | 0.37 | 0.41 | 5.00 | 5.00 | 5.00 | 5.00 | 5.00 |
| Max | 1.00 | 5.00 | 5.00 | 1.07 | 0.77 | 0.94 | 1.35 | 1.03 | 0.97 | 25.00 | 25.00 | 25.00 | 24.00 | 19.00 |
| Skewness | 1.91 | -0.13 | 0.87 | 0.43 | 0.12 | 0.24 | -0.08 | -0.42 | 0.16 | 2.48 | 1.43 | 2.77 | 1.20 | 1.46 |
| Kurtosis | 1.72 | -1.74 | 0.31 | 0.49 | -0.67 | 0.42 | -0.67 | 0.41 | 0.55 | 5.66 | 0.96 | 7.50 | 0.92 | 1.24 |

UAT = unbiased/arcsin transformed.

† $p < 0.1$.

* $p < 0.05$

** $p < 0.01$

*** $p < 0.001$.

## Path model—SEM

The retained model showed excellent fits (Chi-Square = 56.231, df = 63, $p$-value = 0.714; RMSEA = 0.00; CFI = 1.00) and all residual values were close to zero (|value| < 1). All significant paths were in the expected directions (see Fig 2). Controlling for shared variance between types of maltreatment, a history of physical abuse is related to a decreased ability to recognize fear ($\beta$ = -.26, $p$ < .001) and sadness ($\beta$ = -.23, $p$ < .010) in children. Emotional abuse is related to a decreased ability to recognize anger in children ($\beta$ = -.57, $p$ < .001). Sexual abuse is also related to a decreased ability to recognize anger but with a marginal effect ($\beta$ = -.15, $p$ < .100), while emotional neglect is positively associated with the recognition of anger ($\beta$ = .35, $p$ < .050). Physical neglect is associated with less accuracy in recognizing happiness in child facial emotional expressions ($\beta$ = -.25, $p$ < .001).

## Discussion

Our study examined the link between childhood maltreatment and emotion recognition during parenthood. We exposed mothers to child facial expressions, and we examined the

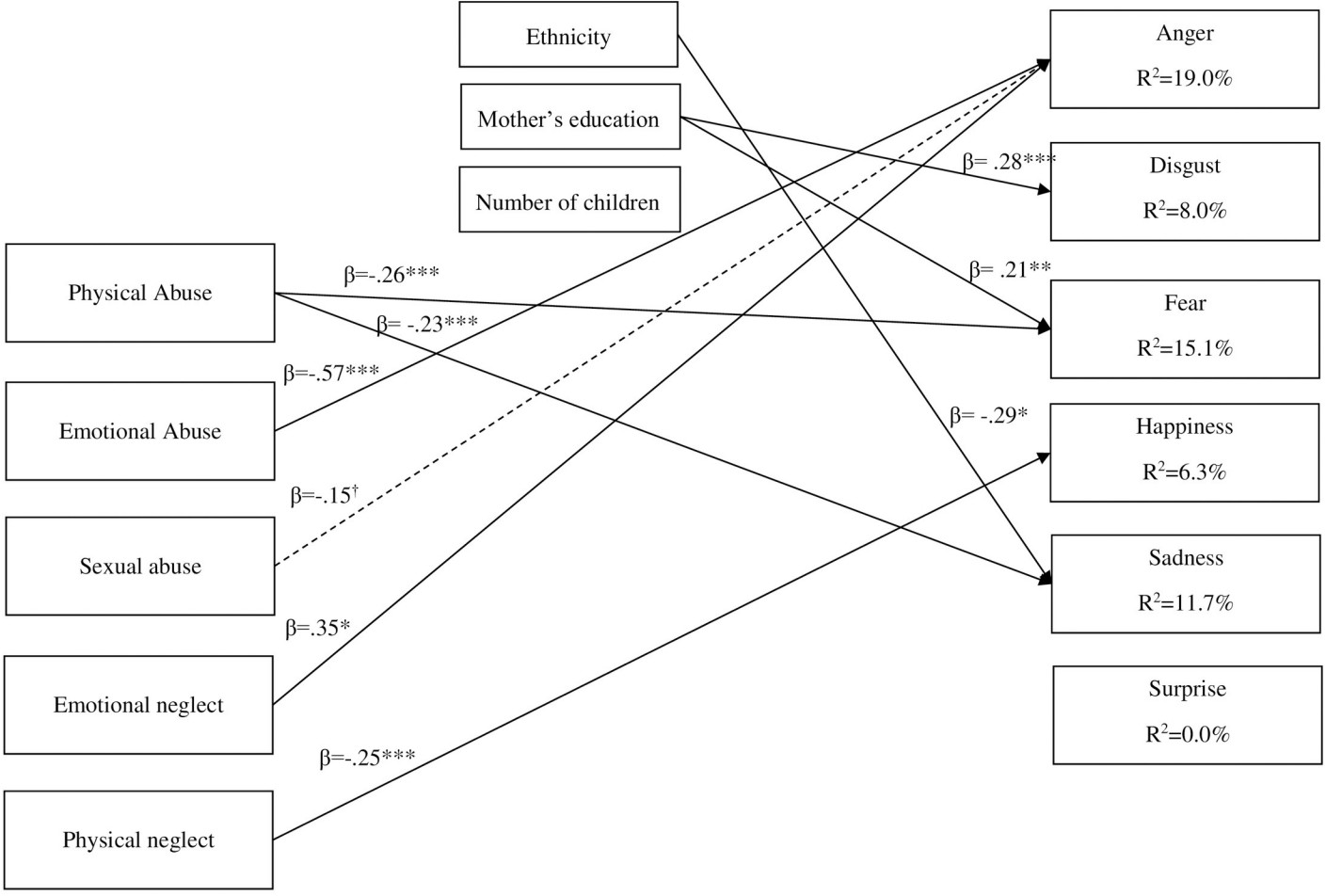

Chi-Square=56.231, df=63, $p$-value=.714; RMSEA=0.00; CFI=1.00
†$p$ < .100; *$p$ < .050; **$p$ < .010; ***$p$ < .001

**Fig 2. Path model results using unbiased hit rates.**

relationship between the six basic emotions and the five most documented forms of maltreatment, which yields new understanding of the differential long-term consequences of childhood maltreatment. Strictly in terms of facial expression recognition, results from the current study suggest that mothers, in general, were less accurate in recognizing facial expressions of disgust and fear, than emotions of anger, happiness, sadness, and surprise. These results align with previous studies in facial expression recognition performed on same-age groups [61]. Although adults are better at recognizing adult faces compared with faces of newborns and children [62], the present study indicates that they keep the same pattern of performance in recognition ability. It is important to note that there are certain interpretive biases in emotion recognition, such as the happy face advantage, which refers to the general tendency of adults to better recognize happiness and to categorize happy faces at a faster rate than negatively-valenced emotions [63, 64].

Regarding the link between childhood maltreatment and emotion recognition, our results indicate that a history of maltreatment influences the ability of mothers to recognize and discriminate the emotions expressed on children's faces. Specifically, we found that a history of physical abuse is related to a lower recognition ability in fear and sadness. Moreover, mothers who experienced emotional abuse in their childhoods show a reduced ability to recognize anger, whereas emotional neglect is related to better accuracy in the detection of anger on children's faces. Finally, mothers with a history of physical neglect are less accurate in recognizing expressions of happiness in children. Our results follow the work of Pollak and his colleagues [34] who revealed that children who have experienced maltreatment recognize emotions differently than non-maltreated controls. Twenty years later, these children may have become parents. We now know that their history of maltreatment will have detrimental effects on their parental role as they will have more difficulty recognizing children's emotions.

Our findings reveal that anger, fear, and sadness are the negative emotions that differentiate those who have a history of childhood maltreatment from those who have not experienced maltreatment during their childhood. Studies have argued that abused children have an enhanced sensitivity to negative facial expressions (i.e., anger, and fear) as a consequence of living in environments where they are exposed to more cases of hostility and threat [37]. Certain authors believe this hypervigilance helps protect children. By identifying signs of anger more rapidly, these children may be able to prevent further abuse [42, 65]. This hypervigilance could decrease as the child becomes an adult and has fewer interactions with his or her maltreating caregivers [46]. Moreover, all the resources maltreated children must dedicate to their survival are allocated to the detriment of their development [66]. Chronic stress experienced by these children affects the development of their brain, particularly the prefrontal cortex and the amygdala, both involved in emotion recognition [67]. Overall, children who have experienced maltreatment show alterations in the neural structures involved in the recognition of facial affect, leading to important deficits in emotional processing [68].

The developmental challenges children growing up in maltreating environments must face could explain a particular developmental trajectory in emotion recognition. As of now, we only have a glimpse of this trajectory. Most research on these children as they reach adulthood has looked at their ability to recognize emotions in adult faces. For example, Hartling and colleagues [69] showed that adults with a history of childhood maltreatment are less accurate in the perception of emotions expressed on adult faces, strictly if they are carriers of a more stress-responsive genetic profile. On the other hand, Gibb and colleagues [45] found that individuals reporting childhood maltreatment are better at perceiving anger in adult faces when presented at lower levels of intensity (morphed at 20 to 40%) but showed similar performances at higher levels of intensity.

Given that deficits in emotion recognition might change between childhood and adulthood, accuracy should also be expected to vary with child and adult facial stimuli. For instance,

Olsavsky and colleagues [70] exposed mothers with and without a history of maltreatment to pictures of infants and adults depicting different emotions. They found a difference in brain activation between both groups, strictly in response to infant faces. Results from our previous work using child facial stimuli revealed that a history of maltreatment not only affects emotion recognition, but also mother-child interactions. Specifically, we found that mothers who had higher performances on the emotion recognition task, combined with a severe history of childhood maltreatment, demonstrated fewer sensitive behaviors towards their child during interactions [51]. Our current findings expand these results by helping us recognize how each form of maltreatment interacts with the way mothers perceive and interpret their child's signals. These findings may lead to the development of specific interventions that address the consequences of each form of maltreatment on emotion recognition. The results of our previous and current studies highlight the specific needs of parents with a history of maltreatment. General emotion recognition interventions may not be appropriate for this population. Rather, our results encourage a more specific approach that focuses on individual difficulties in recognizing and responding to emotions. For instance, Kolijn and colleagues [71] showed that video-feedback, an individualized attachment-based intervention aimed at enhancing the recognition of child emotional expressions and sensitive parenting, may improve mothers' ability to process children's emotions, by reducing the neural effort associated with this task. These results are encouraging as they suggest that the detrimental, long-lasting effects of maltreatment on emotion recognition can be reversed.

## Limitations and future directions

This study is one of the first to document the effect of early maltreatment on mothers' ability to recognize children's emotional expressions. It is also, to our knowledge, the first study to include the six basic emotions in addition to five subtypes of maltreatment. Nevertheless, the findings of the present study must be interpreted in light of certain limitations. First, the relatively small sample size in the current study decreases statistical power. Small effect sizes could be revealed by a larger sample. Therefore, non-significant links should not be interpreted as nonexistent. The small sample size also warrants further studies to ensure the replicability of the present findings. Second, the use of a retrospective self-report instrument to assess mother's exposure to childhood maltreatment may have led certain participants to underestimate or overestimate the prevalence of these experiences. However, the short form of the CTQ is often used in assessing childhood maltreatment experiences in both clinical and nonclinical samples, as it provides a brief, valid, and reliable screening of childhood maltreatment experiences. Third, we conducted a path analysis on cross-sectional data using variables collected at a single time-point. Our results must be interpreted as correlations and do not imply causation. Finally, we examined the relationship between childhood maltreatment and emotion processing in adulthood, without considering other factors such as mental illness. Studies have shown that mental health problems in adulthood, such as major depression or bipolar disorder, can lead to biases in emotion processing [72, 73]. However, Johnson and colleagues [74] found that these biases remained statistically significant after controlling for adult's past and current mental illness.

## Conclusion

Childhood physical abuse, emotional abuse, and physical neglect are related to mothers' decreased ability to recognize facial emotions in children, whereas emotional neglect is related to increased accuracy in the recognition of anger. These findings highlight the need to consider the parent's past experiences when working with families referred to services for childhood

maltreatment. Although several interventions already target these parents, those aimed more specifically at increasing their ability to interpret and respond appropriately to the child signals might not rely on an accurate understanding of the mechanisms underlying this difficulty. Our research provides some answers as to which emotions are more challenging to some parents.

## Supporting information

**S1 File. Database.**
(SAV)

## Acknowledgments

We would like to thank the families who devoted their time to participate in this research, as well as the *Centre de pédiatrie sociale de Gatineau* and the *Carrefour de la Miséricorde* who welcomed us and introduced us to these families.

## Author Contributions

**Conceptualization:** Annie Bérubé, Caroline Blais.

**Data curation:** Jessica Turgeon, Amélie Fournier.

**Formal analysis:** Annie Lemieux.

**Funding acquisition:** Annie Bérubé.

**Investigation:** Annie Bérubé.

**Methodology:** Jessica Turgeon, Annie Bérubé, Caroline Blais.

**Project administration:** Annie Bérubé.

**Resources:** Annie Bérubé.

**Software:** Caroline Blais.

**Supervision:** Jessica Turgeon, Annie Bérubé.

**Validation:** Jessica Turgeon, Annie Bérubé, Caroline Blais, Annie Lemieux.

**Visualization:** Jessica Turgeon.

**Writing – original draft:** Jessica Turgeon.

**Writing – review & editing:** Jessica Turgeon, Annie Bérubé, Caroline Blais, Annie Lemieux, Amélie Fournier.

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
