## [Decision Letter · Decision Letter 0]

22 Sep 2020

PONE-D-20-13201

Recognition of children’s emotional facial expressions among mothers reporting a history of childhood maltreatment

PLOS ONE

Dear Dr. Turgeon,

Thank you for submitting your manuscript to PLOS ONE. After careful consideration, we feel that it has merit but does not fully meet PLOS ONE’s publication criteria as it currently stands. Therefore, we invite you to submit a revised version of the manuscript that addresses the points raised during the review process.

We look forward to receiving your revised manuscript.

Kind regards,

Zezhi Li, Ph.D., M.D.

Academic Editor

PLOS ONE

Journal Requirements:

2. Thank you for including a copy of your related published article and discussing it in your manuscript. Please remove blinded references to this other study and give the full reference for ref. 49 in your manuscript since review is not blinded in this journal.

3. We note that Figure 1 includes an image of a participant in the study. 

4. We note that Figure 1 in your submission contain copyrighted images. All PLOS content is published under the Creative Commons Attribution License (CC BY 4.0), which means that the manuscript, images, and Supporting Information files will be freely available online, and any third party is permitted to access, download, copy, distribute, and use these materials in any way, even commercially, with proper attribution. For more information, see our copyright guidelines: http://journals.plos.org/plosone/s/licenses-and-copyright.

4.1.         You may seek permission from the original copyright holder of Figure(s) [#] to publish the content specifically under the CC BY 4.0 license.

4.2.    If you are unable to obtain permission from the original copyright holder to publish these figures under the CC BY 4.0 license or if the copyright holder’s requirements are incompatible with the CC BY 4.0 license, please either i) remove the figure or ii) supply a replacement figure that complies with the CC BY 4.0 license. Please check copyright information on all replacement figures and update the figure caption with source information. If applicable, please specify in the figure caption text when a figure is similar but not identical to the original image and is therefore for illustrative purposes only.

5. We noted in your submission details that a portion of your manuscript may have been presented or published elsewhere.

"Our manuscript is part of a larger research project that allowed us to publish an article in Child Abuse & Neglect, which we uploaded in the Attach Files. The results related to the manuscript we are submitting to PLOS ONE are original. They have not been published and are not under consideration for publication elsewhere. Please find the details of its specific contributions in our cover letter, in which we explain why our manuscript does not constitute dual publication. "

Please clarify whether thispublication was peer-reviewed and formally published. If this work was previously peer-reviewed and published, in the cover letter please provide the reason that this work does not constitute dual publication and should be included in the current manuscript.

Reviewers' comments:

Reviewer's Responses to Questions

**Comments to the Author**

1. Is the manuscript technically sound, and do the data support the conclusions?

Reviewer #1: Yes

Reviewer #2: Yes

2. Has the statistical analysis been performed appropriately and rigorously? 

Reviewer #1: I Don't Know

Reviewer #2: Yes

3. Have the authors made all data underlying the findings in their manuscript fully available?

Reviewer #1: Yes

Reviewer #2: Yes

4. Is the manuscript presented in an intelligible fashion and written in standard English?

Reviewer #1: Yes

Reviewer #2: Yes

5. Review Comments to the Author

Reviewer #1: Thank you for allowing me to review this interesting manuscript.

Overall, I found it be well written and topical. There is clear attention to detail.

I only have minor comments:

- The topic is important and will be of interest to the readership.

- The manuscript could use some greater detail regarding the implications of this research. What could these results mean for identifying and treating mothers who have experienced trauma and have difficulty with emotion recognition? Are there specific treatments for this deficit?

- The analysis and methodology generally seems appropriate, however, I defer to the other reviewers and editors on this point. Highlight that a path analysis where all variables are collected at a single time-point has limitations.

- It may be worth mentioning the “happy face advantage” (references below) in interpreting some of these results and biases. This is a common bias in facial recognition:

Leppanen, J., Tenhunen, M., & Hietanen, J. (2003). Faster choice reaction times to positive than negative facial expressions: The role of cognitive and motor processes. Emotion, 3, 315–326.

Lipp, O., Craig, B., & Dat, M. (2015). A happy face advantage with male caucasian faces: It depends on the company you keep. Social Psychological and Personality Science, 6, 109–115.

Minor points:

- The first paragraph could possibly be stronger and longer.

- Consider deleting second last sentence of page six: “This paper aims…”

Reviewer #2: The objective of the research entitled " Recognition of children’s emotional facial expressions among mothers reporting a history of childhood maltreatment" is to examine the link between childhood maltreatment and emotion recognition during parenthood. The study is quite interesting and offered new understanding of the differential long-term consequences of childhood maltreatment. I only have a few questions. Below are several suggestions for a revision.

Points to address:

1. Regarding the results of the Paired t-test section, a full list of pair-tested comparisons was presented in Table 1. However, the results state that “mean accuracy was significantly lower for disgust and fear compared with all other emotions” and “anger was the second emotion with the highest accuracy scores, followed by sadness and surprise, respectively”. The t-test table tells which pair comparison is significant, from the t-values and Cohen’s d the reader can infer which emotion have the relative higher or lower accuracy, but it is not straightforward. It would be great if the authors can show the mean accuracy and standard deviation for each emotion in a table or bar plot.

2. Is there a reason why only univariate correlations with p-values smaller than .10 were integrated into the first tested model?

3. For controlling for multiple tests, p-value = .003 was used, I wonder what happens if a different threshold? Are the main results still holds?

4. In line 123, abbreviation CTQ used before first define it.

6. PLOS authors have the option to publish the peer review history of their article (what does this mean?). If published, this will include your full peer review and any attached files.

Reviewer #1: No

Reviewer #2: No

---

## [Author Response · Author response to Decision Letter 0]

22 Oct 2020

Enclosed please find our response to reviewers concerning our manuscript entitled “Recognition of children’s emotional facial expressions among mothers reporting a history of childhood maltreatment”, which we are resubmitting to PLOS ONE with great hopes of achieving publication as an original contribution. 

We first would like to thank the reviewers for their suggestions that have helped clarify and improve the paper. We would also like to thank the editor for giving us the opportunity to resubmit what we believe are important results, as they highlight how childhood maltreatment interferes with mothers’ ability to discriminate child facial expressions of emotion. For clarity, we first state the comments verbatim (in italics), followed by our response. The sections in red indicate changes that were made in the manuscript.

Reviews from the Edition Team

Authors’ reply: We have made the necessary changes to ensure that our manuscript meets PLOS ONE’s style requirements.

2. Thank you for including a copy of your related published article and discussing it in your manuscript. Please remove blinded references to this other study and give the full reference for ref. 49 in your manuscript since review is not blinded in this journal.

Authors’ reply: We have removed blinded references and have added the full reference for ref. 49 in our manuscript (which is now ref. 51, after revisions).

3. We note that Figure 1 includes an image of a participant in the study and 4. We note that Figure 1 in your submission contain copyrighted images.

Authors’ reply: We have removed the Figure from our submission as it is not essential to the reader’s understanding of our paper. The figure provided an illustration of the images to which participants were exposed, but the description in text provides sufficient information and is usually found in other similar papers (Gibb et al., 2009).

5. We noted in your submission details that a portion of your manuscript may have been presented or published elsewhere.

Please clarify whether this publication was peer-reviewed and formally published. If this work was previously peer-reviewed and published, in the cover letter please provide the reason that this work does not constitute dual publication and should be included in the current manuscript.

Authors’ reply: Our publication in Child Abuse & Neglect was peer-reviewed and formally published [https://doi.org/10.1016/j.chiabu.2020.104432]. Our cover letter addresses how our current manuscript does not constitute dual publication, as it extends our previous findings. We have added a sentence in our manuscript to clarify the difference between both manuscripts, see lines 125-128: 

In a previous study, we found that mothers who had higher scores on the Childhood Trauma Questionnaire (CTQ) scales had lower performance scores on the emotion recognition task [51]. Overall scores were created and used in analyses, for both childhood maltreatment experiences and emotion recognition performances. The current study seeks to further explore these results, by drawing conclusions about the differential effects of five maltreatment subtypes on mothers’ ability to recognize six emotions expressed on child faces. To our knowledge, no studies have examined associations between the processing of the six basic facial emotional expressions and the five most documented forms of maltreatment (physical, emotional and sexual abuse, physical and emotional neglect). 

6. Please include captions for your Supporting Information files at the end of your manuscript, and update any in-text citations to match accordingly.

Authors’ reply: We have included captions for our Supporting Information files at the end of our manuscript.

Reviewer #1: 

Thank you for allowing me to review this interesting manuscript.

Overall, I found it be well written and topical. There is clear attention to detail.

I only have minor comments:

- The topic is important and will be of interest to the readership.

Authors’ reply: Thank you, we appreciate this comment.

- The manuscript could use some greater detail regarding the implications of this research. What could these results mean for identifying and treating mothers who have experienced trauma and have difficulty with emotion recognition? Are there specific treatments for this deficit?

Authors’ reply: In response to this comment, we have added sentences in the Discussion section to clarify the implications of our research. The last paragraph now reads:

Our current findings expand these results by helping us recognize how each form of maltreatment interacts with the way mothers perceive and interpret their child’s signals. These findings may lead to the development of specific interventions that address the consequences of each form of maltreatment on emotion recognition. The results of our previous and current studies highlight the specific needs of parents with a history of maltreatment. General emotion recognition interventions may not be appropriate for this population. Rather, our results encourage a more specific approach that focuses on individual difficulties in recognizing and responding to emotions. For instance, Kolijn and colleagues [73] showed that video-feedback, an individualised attachment-based intervention aimed at enhancing the recognition of child emotional expressions and sensitive parenting, may improve mothers’ ability to process children’s emotions, by reducing the neural effort associated with this task. These results are encouraging as they suggest that the detrimental, long-lasting effects of maltreatment on emotion recognition can be reversed.

- The analysis and methodology generally seems appropriate, however, I defer to the other reviewers and editors on this point. Highlight that a path analysis where all variables are collected at a single time-point has limitations.

Authors’ reply: A path analysis on cross-sectional data has limits, particularly regarding data interpretation. Historically, a path analysis was used to test causal effects and therefore a longitudinal data estimate was required. However, this type of analysis is now widely used with cross-sectional data, which makes it possible to discuss correlations between variables in the model. One limitation would be that our conclusions are "correlational" rather than "causal". We have added the sentence below to highlight this limit, see lines 360-362.

Third, we conducted a path analysis on cross-sectional data using variables collected at a single time-point. Our results must be interpreted as correlations and do not imply causation.

- It may be worth mentioning the “happy face advantage” (references below) in interpreting some of these results and biases. This is a common bias in facial recognition:

Leppanen, J., Tenhunen, M., & Hietanen, J. (2003). Faster choice reaction times to positive than negative facial expressions: The role of cognitive and motor processes. Emotion, 3, 315–326.

Lipp, O., Craig, B., & Dat, M. (2015). A happy face advantage with male caucasian faces: It depends on the company you keep. Social Psychological and Personality Science, 6, 109–115.

Authors’ reply: Thank you for this advice. We have mentioned the “happy face advantage” in our manuscript and have added both references to facilitate the interpretation of our results, see lines 288-291.

Minor points:

- The first paragraph could possibly be stronger and longer.

Authors’ reply: Thank you for this advice. We have made changes to the paragraph to make it stronger. The paragraph now reads:

Parental sensitivity refers to a parent’s ability to interpret child signals correctly and to offer an appropriate response [1]. This allows the establishment of a secure attachment relationship [1,2] and promotes the healthy development of young children [3,4]. Among other things, parental sensitivity is associated with positive socioemotional functioning and academic outcomes [5-6]. Conversely, a misinterpretation of the child’s needs or avoidance of caregiving responsibilities leads to important developmental consequences, as the child is unable to use his caregiver as a secure base to explore his environment and seek contact in case of distress [1,7,8].

- Consider deleting second last sentence of page six: “This paper aims…”

Authors’ reply: This sentence was removed.

Reviewer #2: 

The objective of the research entitled " Recognition of children’s emotional facial expressions among mothers reporting a history of childhood maltreatment" is to examine the link between childhood maltreatment and emotion recognition during parenthood. The study is quite interesting and offered new understanding of the differential long-term consequences of childhood maltreatment. I only have a few questions. Below are several suggestions for a revision.

Points to address:

1. Regarding the results of the Paired t-test section, a full list of pair-tested comparisons was presented in Table 1. However, the results state that “mean accuracy was significantly lower for disgust and fear compared with all other emotions” and “anger was the second emotion with the highest accuracy scores, followed by sadness and surprise, respectively”. The t-test table tells which pair comparison is significant, from the t-values and Cohen’s d the reader can infer which emotion have the relative higher or lower accuracy, but it is not straightforward. It would be great if the authors can show the mean accuracy and standard deviation for each emotion in a table or bar plot.

Authors’ reply: In line with this comment, we have added a bar plot with the mean accuracy and standard deviation for each emotion (see fig. 1). 

Fig 1. Mean accuracy and standard deviation for each emotion.

2. Is there a reason why only univariate correlations with p-values smaller than .10 were integrated into the first tested model?

Authors’ reply: This criterion is based on the work of Bendel & Afifi (1977) and Mickey & Greenland (1989) who show that a p-value cut-off of .05 often fails to identify important variables (see references below). We have added this sentence in our manuscript, see lines 215-216.

Bendel, R. B., & Afifi, A. A. (1977). Comparison of stopping rules in forward “stepwise” regression. Journal of the American Statistical association, 72(357), 46-53. https://doi.org/10.1080/01621459.1977.10479905

Mickey, R. M., & Greenland, S. (1989). The impact of confounder selection criteria on effect estimation. American journal of epidemiology, 129(1), 125-137. https://doi.org/10.1093/oxfordjournals.aje.a115101

3. For controlling for multiple tests, p-value = .003 was used, I wonder what happens if a different threshold? Are the main results still holds?

Authors’ reply: The p-value = .003 corresponds to the Bonferroni correction we applied, as mentioned on p.10. Only one paired comparison test has a p-value > 0.003 [Disgust-Fear p=.028]. All other results reach the conservative p-value < 0.003, which we used, and would therefore be retained in the absence of a Bonferroni correction. 

4. In line 123, abbreviation CTQ used before first define it.

Authors’ reply: We have defined the abbreviation before using it. 

We would like to thank both reviewers for their time in reading and reviewing our manuscript.

---

## [Decision Letter · Decision Letter 1]

16 Nov 2020

Recognition of children’s emotional facial expressions among mothers reporting a history of childhood maltreatment

PONE-D-20-13201R1

Dear Dr. Turgeon,

We’re pleased to inform you that your manuscript has been judged scientifically suitable for publication and will be formally accepted for publication once it meets all outstanding technical requirements.

Kind regards,

Zezhi Li, Ph.D., M.D.

Academic Editor

PLOS ONE

Additional Editor Comments (optional):

Reviewers' comments:

Reviewer's Responses to Questions

**Comments to the Author**

1. If the authors have adequately addressed your comments raised in a previous round of review and you feel that this manuscript is now acceptable for publication, you may indicate that here to bypass the “Comments to the Author” section, enter your conflict of interest statement in the “Confidential to Editor” section, and submit your "Accept" recommendation.

Reviewer #1: All comments have been addressed

Reviewer #2: All comments have been addressed

2. Is the manuscript technically sound, and do the data support the conclusions?

Reviewer #1: Yes

Reviewer #2: Yes

3. Has the statistical analysis been performed appropriately and rigorously? 

Reviewer #1: I Don't Know

Reviewer #2: Yes

4. Have the authors made all data underlying the findings in their manuscript fully available?

Reviewer #1: Yes

Reviewer #2: Yes

5. Is the manuscript presented in an intelligible fashion and written in standard English?

Reviewer #1: Yes

Reviewer #2: Yes

6. Review Comments to the Author

Reviewer #1: This paper is well written. The authors have addressed my queries adequately. I believe the paper makes a worthwhile contribution to the literature.

Reviewer #2: The authors have satisfactorily responded to all my questions and made the necessary changes to the manuscript. The revised version of the manuscript appears to be good. I do not have any other questions.

7. PLOS authors have the option to publish the peer review history of their article (what does this mean?). If published, this will include your full peer review and any attached files.

Reviewer #1: No

Reviewer #2: No

---

## [Editor Report · Acceptance letter]

15 Dec 2020

PONE-D-20-13201R1 

Recognition of children’s emotional facial expressions among mothers reporting a history of childhood maltreatment 

Dear Dr. Turgeon:

I'm pleased to inform you that your manuscript has been deemed suitable for publication in PLOS ONE. Congratulations! Your manuscript is now with our production department. 

Kind regards, 

on behalf of

Dr. Zezhi Li 

Academic Editor

PLOS ONE